# Role of Uric Acid in Vascular Remodeling: Cytoskeleton Changes and Migration in VSMCs

**DOI:** 10.3390/ijms24032960

**Published:** 2023-02-03

**Authors:** Elisa Russo, Maria Bertolotto, Valentina Zanetti, Daniela Picciotto, Pasquale Esposito, Federico Carbone, Fabrizio Montecucco, Roberto Pontremoli, Giacomo Garibotto, Francesca Viazzi, Daniela Verzola

**Affiliations:** 1Nephrology and Dialysis Unit, San Luca Hospital, 55100 Lucca, Italy; 2Department of Internal Medicine, University of Genoa, 16132 Genoa, Italy; 3IRCCS Ospedale Policlinico San Martino, 16132 Genoa, Italy

**Keywords:** uric acid, vascular smooth muscle cell, migration, phenotypic transition, vascular remodeling

## Abstract

The mechanisms by which hyperuricemia induces vascular dysfunction and contributes to cardiovascular disease are still debated. Phenotypic transition is a property of vascular smooth muscle cells (VSMCs) involved in organ damage. The aim of this study was to investigate the effects of uric acid (UA) on changes in the VSMC cytoskeleton, cell migration and the signals involved in these processes. MOVAS, a mouse VSMC line, was incubated with 6, 9 and 12 mg/dL of UA, angiotensin receptor blockers (ARBs), proteasome and MEK-inhibitors. Migration property was assessed in a micro-chemotaxis chamber and by phalloidin staining. Changes in cytoskeleton proteins (Smoothelin B (SMTB), alpha-Smooth Muscle Actin (αSMA), Smooth Muscle 22 Alpha (SM22α)), Atrogin-1 and MAPK activation were determined by Western blot, immunostaining and quantitative reverse transcription PCR. UA exposition modified SMT, αSMA and SM22α levels (*p* < 0.05) and significantly upregulated Atrogin-1 and MAPK activation. UA-treated VSMCs showed an increased migratory rate as compared to control cells (*p* < 0.001) and a re-arrangement of F-actin. Probenecid, proteasome inhibition and ARBs prevented the development of dysfunctional VSMC. This study shows, for the first time, that UA-induced cytoskeleton changes determine an increase in VSMC migratory rate, suggesting UA as a key player in vascular remodeling.

## 1. Introduction

Patients with hyperuricemia (HU) are at high risk for developing hypertension [1], as well as cardiovascular (CV) and kidney diseases [2,3,4]. The strength of data from epidemiologic studies proponing UA as a risk factor for CV and total mortality [5,6] encourages the search for the possible pathogenetic mechanisms of this relationship [7,8]. Although the benefits of urate-lowering therapy (ULT) in reducing CV and renal risk in asymptomatic HU are still a matter of debate [9,10,11,12,13], the amount of evidence demonstrating the involvement of UA in kidney and vascular damage [14] supports the hope for valid therapeutic strategies. Recently, HU has been considered as an independent risk factor of vascular inflammation and remodeling [15], and it has been proposed as a promoter of arterial stiffness in humans [16,17,18]. These findings inspired us to investigate the mechanisms triggered by UA in vessel cells. Several experimental studies identified potential pathways underlying UA-induced vascular changes [19]. These mechanisms include inflammation [20,21], oxidative stress [22] and activation of the renin-angiotensin aldosterone system (RAAS) [14], which together contribute to endothelial dysfunction and the proliferation/phenotypic transition of VSMCs, finally resulting in vascular remodeling, arterial stiffness and cardiovascular disease (CVD) [14,18,22].

VSMCs form the medial layer of major arteries and their contractile ability controls the vessel tone. In physiological condition, VSMCs are quiescent but retain remarkable plasticity [23]. On the contrary, in response to injury, VSMCs can become migratory and downregulate components of the smooth muscle-contractile machinery. It has been proposed that these processes, namely the phenotypic switch, lead to vascular remodeling and atherosclerotic lesions [24].

Although several studies have highlighted the importance of UA as a promoter of cell changes [25,26,27], and new mechanisms for UA-mediated vascular injury have recently been provided [28], a better knowledge of the role played by UA, and of the different pathways involved, may be helpful for devising new “cardiovascular protective therapies” aimed to prevent tissue damage.

In this study, we explore the effects of UA on the VSMCs phenotype. With this purpose, we evaluated UA-induced changes in the expression and arrangement of cytoskeleton components and in the migratory capacity in MOVAS cells, an immortalized VSMC line from murine aorta. We also investigated the pathways involved in these processes by pre-treatment of cultured VSMCs with different inhibitors. First, we tested urate transporter blocker (Probenecid) and angiotensin receptor blockers (ARBs), valsartan and losartan. Secondly, bearing in mind that Atrogin-1, a ubiquitin ligase 3 which controls the VSMC phenotype and survival signaling [29], has been previously demonstrated to be upregulated in uremic milieu [30] and to be involved in the proteasome degradation of ubiquitinated proteins in VSMCs [31], we used MG132 (a proteasome inhibitor) to test the involvement of proteasome in UA-induced VSMC alterations. Thirdly, we tested the contribution of MAP kinase in VSMCs migration by U0196, a MEK inhibitor.

## 2. Results

### 2.1. Cell Viability and Area

To examine the effects of UA on cell viability, we treated MOVAS with different UA concentrations (6 to 12 mg/dL) for 24–48 h. Cells showed an increase in proliferation regardless of UA concentration. As graphed in Figure 1A, at 24 h, the percentage of cells rose up to + 11–16% in respect to no treated cells (CTR) (*p* < 0.05) and a similar increment was observed after 48-h of treatment (+8–11%, *p* < 0.05).

After 24–48 h exposure to UA, we found an increase in the cell size, with an overall increase in cell dimension regardless of UA concentration (CTR 627.7 ± 28.3–614 ± 18 (596–545 μm^2^); UA 6 mg/dL 1018 ± 61–1017 ± 100 (974–1010 μm^2^) UA 9 mg/dL 1014 ± 50–1016 ± 110 (841–1010 μm^2^); UA 12 mg/dL 1078 ± 78–905 ± 29 (980–700 μm^2^); median (interquartile range); for each condition *p* < 0.0001 vs. CTR) (Figure 1B). The effects of UA on cell viability and dimensions were independent from concentration and then, for our experimental design, we chose 9 mg/dL UA.

### 2.2. UA Altered α-SMA, Smoothelin B and SM 22α Levels

To test whether UA modifies the cytoskeleton of VSMC, we evaluated alpha smooth muscle actin (αSMA), Smoothelin B (SMT B), and smooth muscle 22α (SM22α) mRNA levels. We observed that after 24 h they were significantly increased by 9 mg/dL UA (α-SMA: +40% vs. CTR, *p* < 0.05; SMT B: +30%, *p* < 0.01; SM 22-α: +240%, *p* < 0.01, *p* < 0.05, respectively) (Figure 2A). On the contrary, after 48-h exposition to UA, cytoskeleton proteins were significantly downregulated (−30/40% vs. CTR, *p* < 0.01) (Figure 2B).

We found the α-SMA protein was upregulated after 24 h UA 9 mg/dL treatment (1.4 fold vs. CTR, *p* < 0.05), but dropped out after 48 h (−40%, *p* < 0.01, Figure 3A). These data were confirmed by immunocytochemistry and immunofluorescence. At 48 h, in UA 9 mg/dL treated cells, immunostaining for α-SMA was fainter than in CTR (Figure 3B) and fluorescence intensity was decreased (−50% vs. CTR, *p* < 0.01) as graphed in Figure 3C. In addition, a deranged architecture and a disorganized orientation of α-SMA were observed (Figure 3D).

### 2.3. UA Promoted MOVAS Cells’ Migration

To explore the role of UA in the phenotypic modification of MOVAS cells, we studied its effect on migration in the micro-chemotaxis chamber. UA stimulated migration independently by time of exposure (Figure 4A). Indeed, incubation with 9 mg/dL UA increased the chemotactic index of cells toward a complete medium (24 h: 1.7–1.9 fold vs. CTR, *p* < 0.0001 48 h: 1.7–2.2 fold vs. CTR, *p* < 0.05). Accordingly, when cells were stained with Alexa-Fluor 594-conjugated phalloidin, UA induced the F-actin re-arrangement in thinner and poorly oriented fibers localized at cortical level, while in CTR, its compact polymerization in stress fibers along the major cell axis was observed (Figure 4B). These effects were blunted by 10–20 μm Probenecid (Appendix A.

### 2.4. UA Promoted MOVAS Cells Migration through the p44/42 MAPK Pathway

Figure 5A,B show the effects of 9 mg/dL UA on p44/42 MAPK phosphorylation and its involvement in migratory capacity. UA treatment raised the levels of phosphorylation until ~2–2.5 folds over a 6-h incubation period (*p* < 0.05 vs. T0). As a next step, we observed that UA-induced migration was stopped by the MEK-1 and -2 inhibitor U0196 (Figure 5C,D).

### 2.5. UA Induced Atrogin-1 Expression

In an attempt to identify a possible mechanism by which UA modifies VSMC phenotype, we evaluated the involvement of Atrogin-1, an E3 ubiquitin-ligase that directs the poly-ubiquitination of proteins, on the expression of α-SMA. First of all, we found out that at 48 h, 9 mg/dL, UA increased Atrogin-1 gene expression -2 fold vs. CTR (*p* < 0.05) (Figure 6A). In addition, when we measured fluorescence intensity in cells exposed to 9 mg/dL UA and immuno-stained with antibody to Atrogin-1, we found it increased 2.75-fold vs. no treated cells (*p* < 0.05) (Figure 6B,C).

Then, we performed co-localization studies of Atrogin 1 and α-SMA and we observed that they were oppositely expressed: in CTR, α-SMA was highly expressed (green) and Atrogin-1 (red) was faintly expressed, whereas in UA-treated cells, the red fluorescence was predominant (Figure 6C). Probenecid treatment blunted these effects (Appendix A).

Finally, to confirm the role of the ubiquitin–proteasome system (UPS) in VSMC phenotypic modification, we pretreated MOVAS cells with 5 µm MG132 which, at 24 h, inhibited the overexpression of α-SMA, SMT B and SM 22 α (*p* < 0.01–0.001 vs. UA) (Figure 7A) and rescued the migratory capacity induced by UA (Figure 7B).

### 2.6. AT1 Receptor Blockers (ARBs) Blunted VSMC Modifications Induced by UA

We examined whether Losartan (L) and Valsartan (V) could prevent the development of the dysfunctional VSMC phenotype induced by UA. As depicted in Figure 8A, L and V treatment inhibited hypertrophy induced by UA and cells had areas similar to CTR. Moreover, we observed that L and V downregulated α-SMA, SMT B and SM 22 α mRNA levels induced by UA (*p* 0.05–0.0001 vs. UA treated cells) (Figure 8B–D).

However, L and V had an inhibitory effect on UA-induced migration (Figure 9A) and this was confirmed by phalloidin staining that was similarly distributed in CTR in L- and V-treated cells, (Figure 9B). Furthermore, L and V blunted p44/42 MAPK phosphorylation induced by UA (Figure 10A,B).

## 3. Discussion

Several novel findings were made in this study: (1) UA treatment induced structural and functional transition into a proliferative/migratory phenotype of VSMCs; (2) UA determined an increase in the cellular migration rate of VSMCs; (3) Blocking the internalization of UA and the inhibition of proteasome, RAAS or MEK activity abolished the UA-induced cellular changes in VSMCs and their increase in migration rate. The mechanisms by which UA induces VSMCs changes are summarized in Figure 11. Taken together, these results provide compelling evidence that UA is a key player in vascular remodeling by proteasome and RAAS activation.

In human cells, UA has emerged as a major cause of damage in different pathophysiological conditions [21,25,26,27]. We found that increasing UA promoted the proliferation of VSMCs (Figure 1A) and determined an increase in cell area (Figure 1B) in murine cells. Our results confirm [32] and expand previous findings corroborating the hypothesis that UA-induced VSMC hypertrophy is implicated in the development and progression of vascular damage observed in patients with increased UA levels [18,33].

VSMCs mainly has a contractility phenotype, which is responsible for the modulation of vascular tone. They also possess extensive plasticity such that they can be stimulated to transform from a contractile state into proliferative/migratory state [34]. The mature contractile phenotype of VSMCs is morphologically characterized by low numbers of protein synthesis structures, and a high expression of proteins involved in muscle contraction and anchorage including αSMA [35], smooth SM22α [36] and SMT B [37]. Synthetic phenotypes are associated with abnormal mechanical forces, which lead to tissue repair and remodeling. With this in mind, we analyzed the changes in cytoskeleton protein expression induced by UA. The protein levels of α-SMA, SM22 α and SMT B, after a transitory growth in the earlier hours, were significantly decreased after 48 h of exposure to UA (Figure 2), sustaining the phenotypic switch to a synthetic pathway. The implication of UA in cytoskeleton changes of VSMC is further substantiated by our findings of a derangement of α-SMA architecture and a disorganized orientation of actin fibers in UA treated cells, compared to controls (Figure 3).

The emerging role of cell migration as a necessary process in vessel wall remodeling has stimulated strong interest in cellular events and molecular mechanisms of VSMC migration. The migratory–proliferative phenotype is characterized by certain VSMC cytoskeletal structural changes that enable it to migrate to the source of initiating stimuli [38]. For the first time, our study demonstrated UA as a potential trigger for these changes (Figure 4A). Cell migration begins with the stimulation of cell surface receptors that transduce the external signal to a series of coordinated remodeling events finally altering the structure of the cytoskeleton. Early signaling events provoke actin polymerization, which is critical for generating membrane protrusions at the leading-edge during cell migration [39]. In the present study, UA-treated cells showed cytoskeleton remodeling and rearrangement of F-actin fibers (marked with phalloidin staining), probably causing a detachment of focal contacts at the trailing edge, propelling the cell toward the stimulus (Figure 4B). These results are of most interest because of the known mechanism by which VSMCs undergoing confined migration may promote the secretory phenotype and further accelerate vascular ageing [40]. The inhibition by Probenecid of this cascade of events, highlighted as the internalization of UA, is essential for the definition of VSMCs’ structure (Appendix A).

Much research exists about the role of biochemical signaling in VSMC migration, which remains challenging for our limited knowledge of biology complexity. Major goals of these studies include the definition of the proximal signals (i.e., UA in our study), as well as the signal transduction pathways that affect cell movement. Intracellular signal-transduction pathways such as mitogen-activated protein kinase/extracellular signal-regulated kinase (MAPK p44/42), leading to VSMC proliferation and migration have been demonstrated to be activated by several factors [41,42]. Our study identified UA as one of the potential triggers for these processes (Figure 5).

The UPS is central for protein turnover as the main pathway for the degradation of proteins related to cell-cycle regulation, signaling and differentiation in eukaryotic cells, including VSMC [43]. Ubiquitination of the target protein is a reversible post-translational modification, which can occur under the action of ubiquitin–ligase enzymes [44]. Several studies suggested that UPS promotes protein degradation, underlying transition from contractile to proliferative VSMC phenotype, with the proteasome inhibitor shifting VSMC away from a synthetic morphology [45]. Our study reveals that Atrogin-1, a cardiac E3 ligase, was two-fold up regulated in VSMCs treated with 9 mg/dL UA compared to no treated cells (Figure 6). Moreover, a pre-treatment with MG 132, a proteasome inhibitor, prevented all cytoskeleton and migration property changes induced by UA (Figure 7). These findings suggest that in cells exposed to increased UA, the UPS is a strong effector of VSMC phenotypic changes, provoking transition towards a synthetic VSMC phenotype as happened during the development of atherosclerosis and vascular diseases.

Angiotensin II (Ang II) plays a vital role in arterial remodeling through multiple pathways [46]. It accentuates the inflammatory response to the stimuli and promotes the migratory–proliferative activity of VSMC [47]. The present study contributes to the comprehension of the protective mechanisms exerted by RAAS inhibition. We described how ARB treatment (valsartan and losartan 10 µM) inhibited changes in cytoskeleton components (Figure 8), protein overexpression of p44/42 MAPK (Figure 10) and migration in UA-stimulated VSMCs (Figure 9), indicating a possible mechanism by which ARBs were shown to be able to reduce neointimal hyperplasia.

This study has some limitations. A shift in cell phenotype is observed when VSMCs are removed from their native environment and placed in a culture, presumably due to the absence of the physiological signals that maintain and regulate the VSMC phenotype. Nevertheless, the inhibition both of UPS and RAAS (Figure 7 and Figure 9), significantly blunts the changes in cytoskeleton proteins and cell phenotype we observed, suggesting UA as the through effector of this set of pathologic modifications. Moreover, the experiments were not performed using primary cells. Therefore, the results of the present study need to be confirmed by experiments on primary cell culture and on animal models.

## 4. Materials and Methods

The procedures were in accordance with the Declaration of Helsinki. The study was approved by the Ethical Committee of Regione Liguria (Comitato Etico Regionale, CER) on 28 June 2021 (Protocol UA-MSTN-VASC; N. Registro CER Liguria: 129/2021–DB id 11314).

### 4.1. Cell Culture

MOVAS cells, an immortalized smooth-muscle cell line from murine aorta, were obtained from ATCC (CRL-2797™). Cells were grown in DMEM (Euroclone S.p.A., Milan, Italy) medium supplemented with 10% (*v/v*) FBS, 0.2 mg/mL G-418 (Euroclone S.p.A.). Cells were grown at 37 °C in a humidified 5% CO_2_ conditions.

### 4.2. Cell Treatments

MOVAS were grown to sub-confluence and then cultured in starving medium (with 1% FBS) in the presence of Uric Acid (UA) (6–12 mg/dL) (Merck group, Milan, Italy) for 24–48 h. Control cells (CTR) were not treated with UA. Firstly, we evaluated the effects of different concentrations of UA on cell viability and then, on the cell dimensions, the migratory capability and the expression of cytoskeleton proteins (α-SMA, SMT B, SM22α). Furthermore, the role of ubiquitin–proteasome (UPS) on UA effects was studied, evaluating atrogin-1 (a F-box protein or Ubiquitin-protein ligase E3) expression and exposing cells to MG132 (5 µM) (Medchemexpress, DBA Italia s.r.l Seregno, Italy), a potent, reversible and cell-permeable proteasome inhibitor. Probenecid (5–10 µM) (Merck group) was added to the cell culture to test its capability to blunt migration and atrogin-1 expression.

To understand if UA effects were direct or mediated by RAS activation, Losartan (L) or Valsartan (V) (both 10 µM) (Medchemexpress, DBA Italia s.r.l.) were added 1 h earlier than 9 mg/dL AU. Lastly, the UA-induced activation of p44/42 MAPK was studied and its effects on cell migration through pretreatment with U0126 (10 µM) (Merck group).

### 4.3. MTT Assay

This assay for cell viability is based on the reduction of 3-(4.5-dimethylthiazol-2-yl)-2.5-diphenyltetrazolium bromide (MTT) (Merck Group) by mitochondrial dehydrogenase in viable cells to produce a purple formazan product. Each experiment was performed according to the protocol as previously described [27].

### 4.4. mRNA Analysis

MOVAS cells were incubated for 24–48 h with or without 9 mg/dL UA. Total RNA was isolated using the Tri-Reagent (Zymo Research, Aurogene, Rome, Italy). A total of 1 µg RNA was used for cDNA synthesis with SCRIPT RT Supermix (BioRad Laboratories, Segrate, Italy). PCR amplification was carried out in a total volume of 10 µL, containing 1 µL cDNA solution, 5 µL SensiFAST SYBR no ROX mix (Meridian Bioscience, Aurogene, Rome, Italy), 0.08 µL of each primer (Tib Molbiol, Genoa, Italy) and 3.84 µL of nuclease-free water. β-actin was quantified, and used for the normalization of expression values of the other genes.

Assays were run in triplicate on MasterCycler realplex PCR system (Eppendorf, Hamburg, Germany). The primer sequences are reported in Table 1.

### 4.5. Western Blot Analysis

The cell layers were lysed in cold buffer (20 mM HEPES, 150 mM NaCl, 10% (*v*/*v*) glycerol, 0.5% (*v*/*v*) NP-40, 1 mM EDTA) supplemented with Protease inhibitor cocktail (Merck Group). Protein concentration was determined by using the Bicinchonic Protein assay kit (Euroclone S.p.A.) and 20–30 µg was resolved on SDS-polyacrylamide gels and electro-transferred to Amersham™ Hybond™ PVDF membrane (Euroclone S.P.A.). Blots were incubated o.n. at 4 °C with monoclonal anti α-smooth muscle actin (α-SMA) antibody (clone1A4) (Merck group), re-probed with anti H3 Histone antibody (Santa Cruz Biotechnology, DBA Italia s.r.l),) or p44/42 MAPK (R&D System, Bio-Techne s.r.l., Milan, Italy) and then incubated in horseradish peroxidase secondary antibodies (Cell Signaling Technology, Euroclone s.p.a., Milan, Italy). Immunoblots were developed with Immobilon Western chemiluminescent HRP substrate (Merck Group) and band intensities were determined using Alliance imaging system (Uvitec, Cambridge, UK).

### 4.6. Immunocytochemistry and Immunofluorescence

MOVAS grown on chamber slides were incubated for 24–48 h with or without UA (9 mg/dL). After a five minute incubation in cold methanol, cells were exposed to anti α-SMA (Merck Group) and/or anti-atrogin-1 (rabbit polyclonal antibody) (ECM Bioscences, DBA Italia s.r.l.). For immunocytochemistry, cells were probed with an ultra-polymer goat anti-mouse IgG conjugated to HRP (ImmunoReagents Inc, Microtech, Pozzuoli, Italy) and developed with DAB detection kit (Roche, Microtech). Slides were counterstained with hematoxylin and examined by light microscopy.

For immunofluorescence, secondary antibodies conjugated to Alexa Fluor^tm^ 488 and/or 594 were used (Invitrogen, Thermo Fisher Scientific, Monza, Italy). Nuclei were stained with DAPI.

The levels of cell fluorescence were determined by ImageJ and they were calculated as corrected total cell fluorescence (CTCF) using this formula: CTCF = Integrated Density–(Area of selected cell × Mean fluorescence of background readings) (https://theolb.readthedocs.io/en/latest/index.html, accessed on 20 August 2021).

### 4.7. MOVAS Cell Dimension Analysis

The area of MOVAS cells was evaluated after 24 and 48 h exposure to UA (6–12 mg/dL) and compared with that of CTR. In brief, cells grown on chamber slides (EZ chamber slides; Merck Group) were fixed with 2% paraformaldehyde, stained with Hematoxylin–Eosin and photographed with a Leica Microsystems microscope (GmbH Wetzlar, Germany) (40× magnification). Images were then analyzed using MetaMorph^®^ NX software (Molecular Devices): the areas of randomly selected single cells were defined using the cursor and automatically estimated. The median values were calculated upon measurement of 100 cells/condition.

### 4.8. F-Actin Detection by Fluorescence Microscopy

Cells grown on EZ- chamber slides (Merck group) were exposed to UA (9 mg/dL) with or without inhibitors for 24–48 h. Cells were fixed with 2% paraformaldehyde, permeabilized with 0.05%Triton × 100 and stained with 5 U/mL Alexa-Fluor 594-conjugated phalloidin (ThermoFisher Scientific, Monza, MB, Italy). Nuclei were counterstained with DAPI. Cells were washed three times with PBS and analyzed by fluorescent microscopy.

### 4.9. Migration Assays

MOVAS were incubated for 24–48 h in the presence or absence of UA 9 mg/dL and/or Probenecid, Losartan, Valsartan and MG-132. Then, cells were tested for migration in a micro-chemotaxis Boyden chamber (Neuro Probe Inc., Gaithersburg, MD, USA) using a 8 μm-pore size, polycarbonate polyvinylpyrrolidone-free filters (Millipore). The lower wells of the chemotaxis chambers were filled with medium alone. After incubation (3 h, 37 °C), the filters were removed from the chambers, fixed and stained with May Grunwald–Giemsa stain (Carlo Erba, Cornaredo, Italy). Each condition was performed in duplicate. The cells of five random oil-immersion fields were counted and the chemotaxis index was calculated from the number of cells migrated to the test samples divided by the number of cells migrated to the control [12].

### 4.10. Statistical Analysis

Data are given as mean ± SEM. Statistical analysis was performed by Student’s *t*-test or one-way Anova with Bonferroni’s post-test for multiple group comparison. Statistical significance was set at *p* < 0.05. All statistical analyses were performed using Graph Pad Prism version 5.00 for Windows, GraphPad Software, San Diego, CA, USA.

## 5. Conclusions

We found that even a slight increase in UA levels promotes VSMCs’ proliferation and their shift towards synthetic phenotype through proteasome and RAAS activation. These results provide the evidence of the cellular mechanisms that induce those vascular remodelings and damage [17] which, when they become clinically significant, underlie the increased CV and renal risk of patients with asymptomatic HU.

Furthermore, our results suggest a novel role of UPS which can modulate cytoskeleton proteins in UA-stimulated VSMCs. UPS-signaling inhibition might serve as a therapeutic target in the management of vascular damage, and RAAS inhibition might be thought to prevent UA-induced atherosclerosis even in the absence of hypertension.

## Figures and Tables

**Figure 1 ijms-24-02960-f001:**
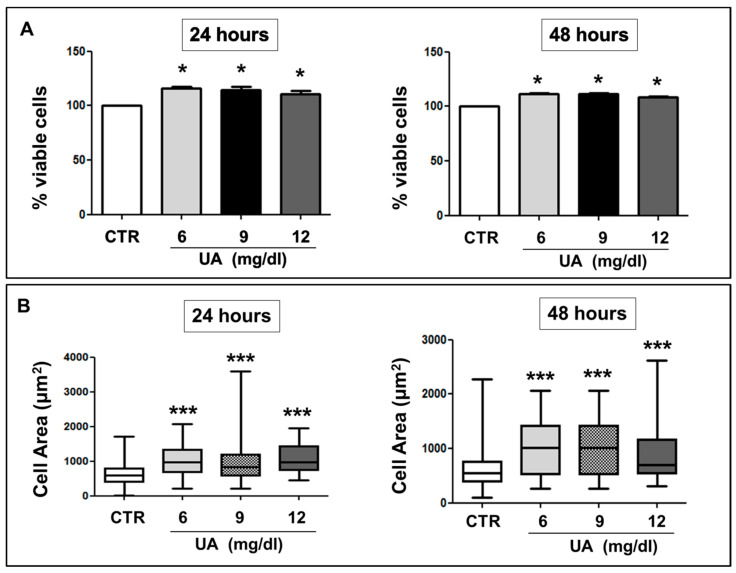
UA effects on cell viability and MOVAS cells area. (**A**) Incubation of VSMCs with UA for 24 and 48 h stimulated proliferation at concentrations of 6, 9 and 12 mg/dL and (**B**) a significant increase in VSMC area, regardless of UA concentration and time exposition. All results represent means ± SEM (**A**) and median ± IQR (**B**) obtained from three/four independent experiments. * *p* < 0.05, *** *p* < 0.0001 vs. CTR. VSMC = Vascular Smooth Muscle Cells; UA = Uric Acid; CTR = controls, no treated cells; IQR = Interquartile Range.

**Figure 2 ijms-24-02960-f002:**
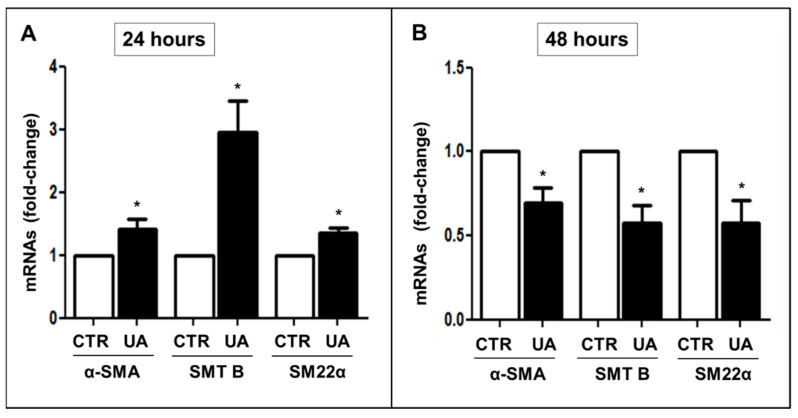
UA effect on cytoskeleton components of MOVAS cells. (**A**) 24 h 9 mg/dL UA exposition increased α-SMA, SMT B and SM22α mRNAs, (**B**) whereas at 48 h they were down regulated. mRNAs were evaluated by rt-PCR. All results represent means  ±  SEM obtained from four/five independent experiments and are expressed as fold change to CTR. * *p*  <  0.05 vs. CTR. UA  =  Uric Acid; α-SMA = α-Smooth Muscle Actin; SMT B = Smoothelin B; SM22α = Smooth Muscle 22α; rt-PCR = real time PCR; CTR = Control, untreated cells.

**Figure 3 ijms-24-02960-f003:**
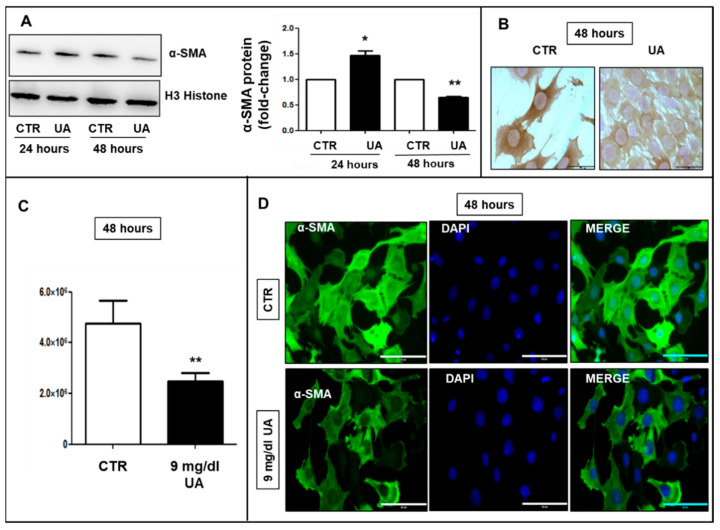
A total of 9 mg/dL UA promoted α-SMA re-arrangement in MOVAS cells. (**A**) 24-h UA exposition rose up α-SMA levels, which fell down at 48 h, as shown by Western blot analysis. Blots were stripped and reproofed with antibody to anti-H3 Histone. (**B**) Evaluation of α-SMA by immunocytochemistry and (**C**,**D**) immunofluorescence confirmed the downregulation after 48 h UA treatment. (**D**) In addition, α-SMA (green) network in UA-treated cells was deranged with respect to CTR. All results represent means  ±  SEM obtained from three/four independent experiments and are expressed as fold change to CTR. * *p*  <  0.05; ** *p* < 0.01 vs. CTR. CTCF was calculated by Image J and for condition at least 100 cells were evaluated from three independent experiments. (**B**) = Magnification ×630 and (**D**) = Magnification ×400, scale bar = 50 μm. UA  =  Uric Acid; α-SMA = α-Smooth Muscle Actin; CTR = Control, untreated cells; CTCF = corrected total cell fluorescence; DAPI = 4′,6-diamidino-2-phenylindole (blue).

**Figure 4 ijms-24-02960-f004:**
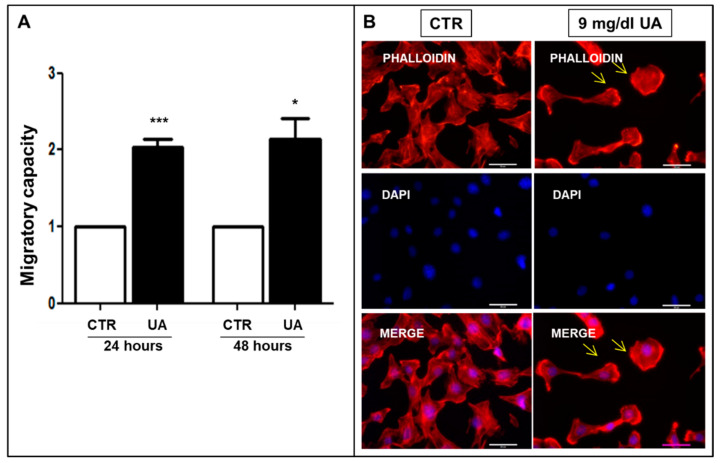
UA triggers F-actin different polymerization in MOVAS cells. (**A**) UA induced greater migratory capacity and (**B**) caused F-actin stress fiber organization, visualized with AlexaFluor 594-Phalloidin staining. Arrows point out cortical F-actin distribution. All results represent means  ±  SEM obtained from four/eight independent experiments and are expressed as fold change to CTR. * *p*  <  0.05; *** *p* < 0.0001 vs. CTR. Magnification ×400, scale bar = 50 μm. UA  =  Uric Acid; VSMC = Vascular Smooth Muscle Cells; CTR = Control, untreated cells; DAPI = 4′,6-diamidino-2-phenylindole.

**Figure 5 ijms-24-02960-f005:**
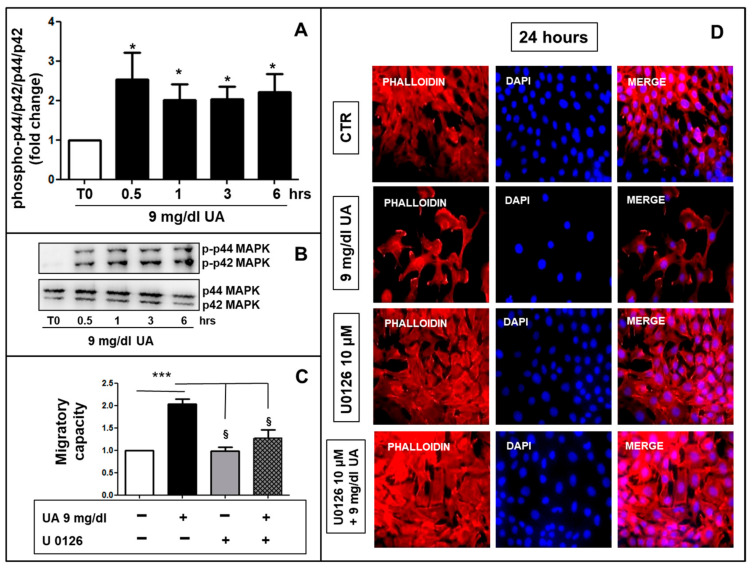
UA induced a time dependent p44/42 MAPK phosphorylation. (**A**) 9 mg/dL UA induced p44/42 MAPK phosphorylation (* *p* < 0.05 vs. T0), (**B**) as shown by Western blot analysis. (**C**) U0196 reversed migratory capacity as well as cortical F-actin distribution induced by UA (* *p* < 0.05 vs. T0, *** *p* < 0.0001 vs. CTR, § *p* < 0.05 vs. UA treated cells) (**D**) as visualized by AlexaFluor 594-Phalloidin staining. Magnification = ×200, scale bar = 30 μm. All results represent means  ±  SEM obtained from three/four independent experiments and are expressed as fold change to T0 (**A**,**B**) or CTR (**C**,**D**). UA = Uric Acid; CTR = Control, untreated cells; MAPK = Mitogen-activated protein kinase; T0 = Time 0; hrs. = hours; DAPI = 4′,6-diamidino-2-phenylindole.

**Figure 6 ijms-24-02960-f006:**
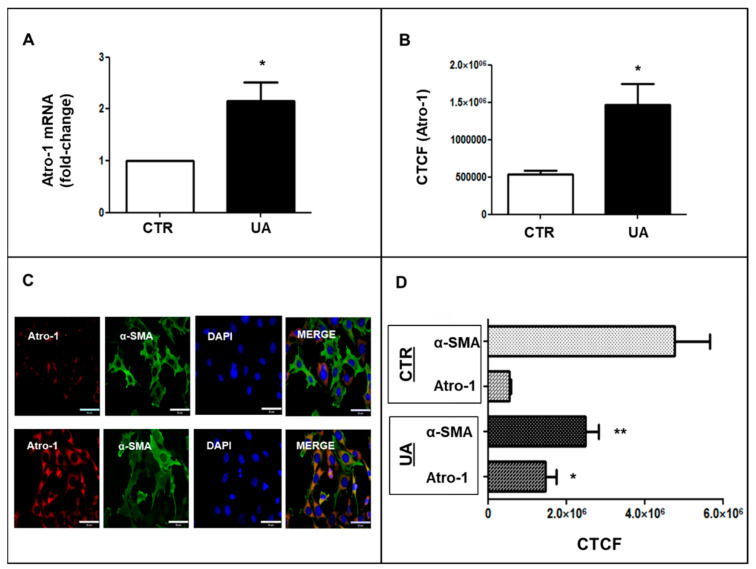
Atrogin-1 (red) and αSMA (green) after 48 h of treatment with UA 9 mg/dL. (**A**) UA treatment upregulated Atrogin-1 mRNA and (**B**) protein expression evaluated by immunofluorescence and reported as CTCF. mRNA was detected by rt-PCR. (**C**) UA-induced changes in expression and distribution of Atrogin-1 (red) and α-SMA (green). (**D**) Signal intensities were evaluated by Image J and expressed as CTCF. * *p* < 0.05 and ** *p* < 0.01 vs. CTR. rt-PCR results represent means  ±  SEM obtained from three/four independent experiments and are expressed as fold change to CTR. For CTCF, at least 100 cells for condition from three independent experiments were evaluated. Magnification = ×400, scale bar = 35 μm. Atro-1 = Atrogin-1; α-SMA = α-Smooth Muscle Actin; VSMC = Vascular Smooth Muscle cells; UA = Uric Acid; rt-PCR = real time PCR; CTR = control untreated cells; CTCF = Corrected Total Cell Fluorescence; DAPI = 4′,6-diamidino-2-phenylindole (blue).

**Figure 7 ijms-24-02960-f007:**
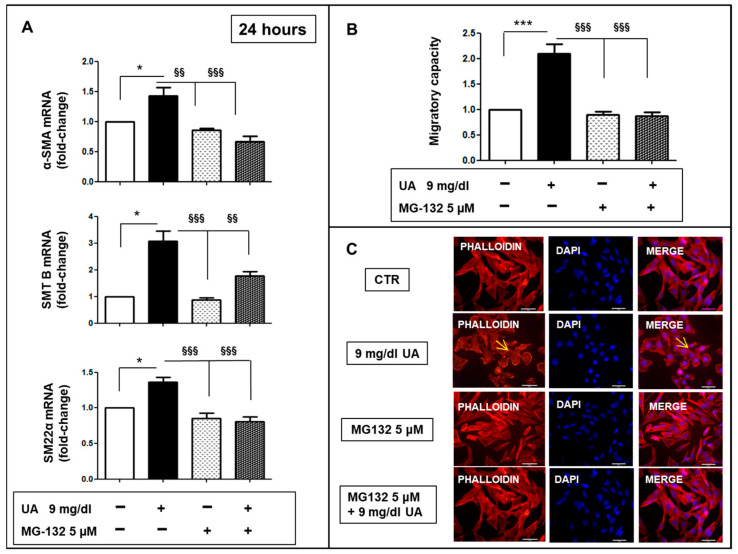
Role of ubiquitin–proteasome system in MOVAS phenotypic modification. (**A**) 5 μM MG 132 pre-treatment inhibited the UA mediated overexpression of α-SMA, SMT B and SM22 α mRNAs, evaluated by rt-PCR, as well as (**B**) the migratory capacity. (**C**) AlexaFluor 594-Phalloidin staining showed that only in UA treated cells, F-actin has a cortical distribution, as pointed by yellow arrows (magnification ×400, scale bar = 50 μm). Results represent means  ±  SEM obtained from three/four independent experiments and are expressed as fold change to CTR. * *p* < 0.05 and *** *p* < 0.0001 vs. CTR; §§ *p* < 0.01, §§§ *p* < 0.0001 vs. UA treated cells. UA  =  Uric Acid; α-SMA = α-Smooth Muscle Actin; SMT B = Smoothelin B; SM22α = Smooth Muscle 22α; CTR = Control; DAPI = 4′,6-diamidino-2-phenylindole.

**Figure 8 ijms-24-02960-f008:**
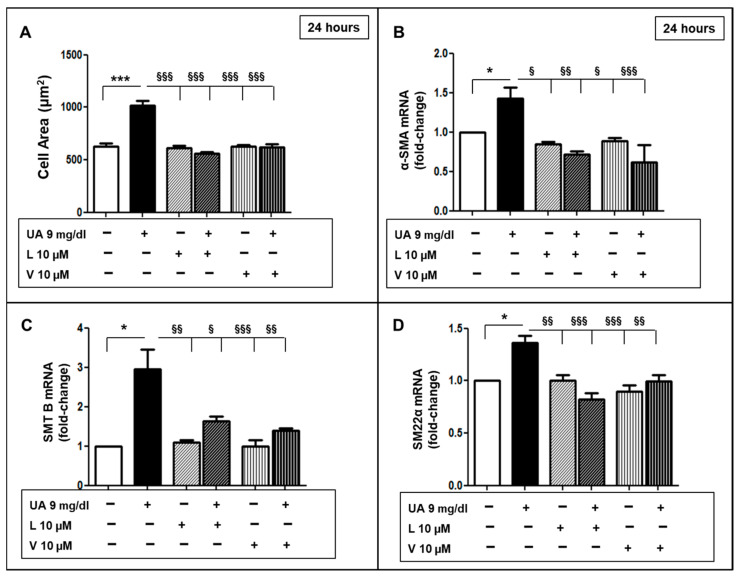
ARBs blunted UA-mediated VSMC phenotypic modification. (**A**) Both L and V (both 10 μM) inhibited the cell area increase induced by UA. (**B**–**D**) L and V treatment protected VSMCs from α-SMA, SMTB and SM 22 α mRNA changes induced by UA. mRNA levels were measured by rt-PCR. Results represent means  ±  SEM obtained from three/four independent experiments and are expressed as fold change to CTR. * *p* < 0.05 and *** *p* < 0.0001 vs. CTR; § *p* < 0.05, §§ *p* < 0.01, §§§ *p* < 0.0001 vs. UA treated cells. ARBs = Angiotensin Receptor Blockers; VSMC = Vascular Smooth Muscle Cells; L = Losartan, V = Valsartan; UA = Uric Acid; α-SMA = α-Smooth muscle actin; SMT B = Smoothelin B; SM22α = Smooth Muscle 22α; rt-PCR = real time PCR; CTR = control, untreated cells.

**Figure 9 ijms-24-02960-f009:**
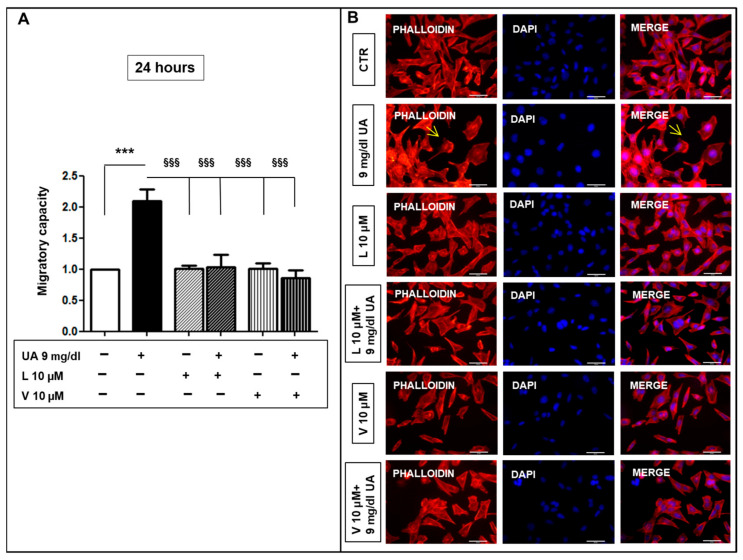
ARBs inhibited VSMC migratory capacity induced by UA. (**A**) Graph represents the inhibitory effects of L and V (both 10 μM) on cell migration after 24 h exposition to 9 mg/dL UA. (**B**) This effect was confirmed by AlexaFluor 594-Phalloidin staining. As pointed by yellow arrows, only in UA-treated MOVAS, F-actin had a cortical distribution (magnification ×400, scale bar = 50 μm). Results represent means  ±  SEM obtained from three/four independent experiments and are expressed as fold change to CTR. *** *p* < 0.0001 vs. CTR; §§§ *p* < 0.0001 vs. UA treated cells. ARBs = Angiotensin Receptor Blockers; VSMC = Vascular Smooth Muscle Cells; UA = Uric Acid; L = Losartan, V = Valsartan; CTR = control, untreated cells.

**Figure 10 ijms-24-02960-f010:**
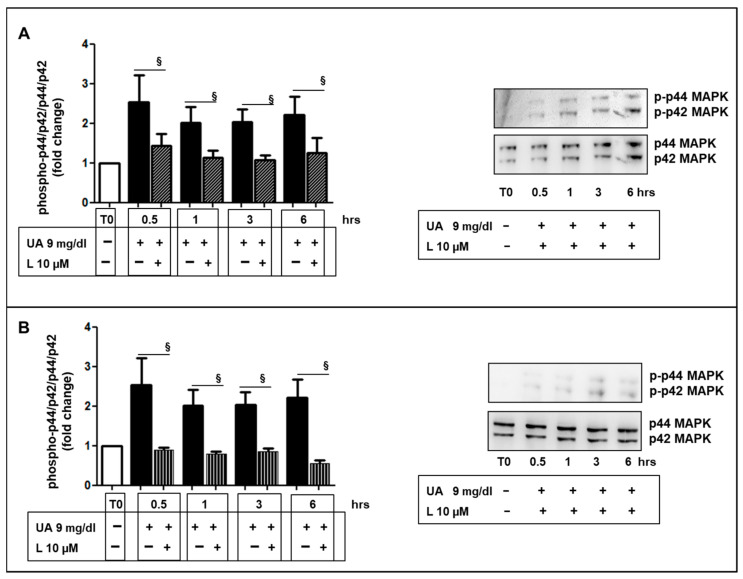
ARBs inhibited UA-mediated p44/42 MAPK activation. (**A**) L and (**B**) V inhibited p44/42 MAPK phosphorylation in MOVAS exposed to UA 9 mg/dL. The graphs represent relative phospho- p44/42 MAPK protein abundance normalized to p44/42 MAPK and data are expressed as fold change with respect to T0 and significance is expressed vs. UA treated cells at the same time and as means ± SEM of three independent experiments. § *p* < 0.05 vs. UA. ARBs = Angiotensin Receptor Blockers; L = Losartan; V = Valsartan; UA = Uric Acid; MAPK = Mitogen-activated protein kinase;T0 = Time 0; hrs. = hours.

**Figure 11 ijms-24-02960-f011:**
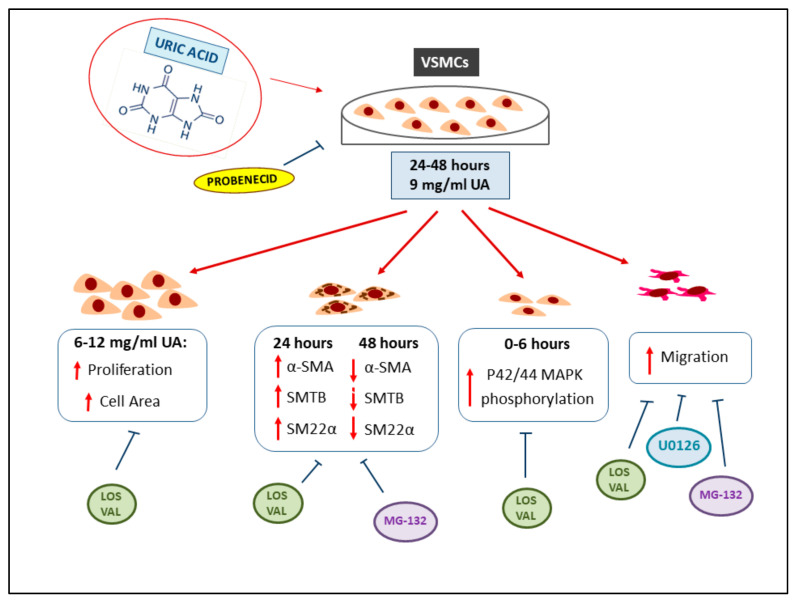
UA effects on VSMCs. The 24–48 h incubation of VSMCs with UA results in the increased area and proliferation of cells, changes in cytoskeleton protein expression, increased p42/44 MAPK phosphorylation and migration. The inhibitors used were able to block these processes induced by UA. Abbreviations: VSMCs = Vascular Smooth Muscle Cells; UA = Uric Acid; α-SMA = α-Smooth muscle actin; SMT B = Smoothelin B; SM22α = Smooth Muscle 22α; MAPK = Mitogen-activated protein kinase; LOS = Losartan; VAL = Valsartan; MG-132 = proteasome inhibitor; U0126 = MEK inhibitor.

**Table 1 ijms-24-02960-t001:** Sequences of gene-specific primers used in qRT-PCR.

Primers	Forward	Reverse
Mouse α-SMA	tcctgacgctgaagtatccgat	ggccacacgaagctcgttatag
Mouse SMT-B	aactggctacactctcaacagcga	aaggtggcagccttaatctcctga
Mouse SM22 α	cggcagatcatcagttagaaag	gggctgaggctgaggataggt
Mouse Atrogin-1	gaggcagattcgcaagcgtttgat	tccaggagagaatgtggcagtgtt
Mouse β-Actin	catcactattggcaacgagcg	atggatgccacaggattcca

## Data Availability

The authors confirm that all data underlying the findings are fully available without restriction. All relevant data are within the paper.

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
