# Peer review of "Role of Uric Acid in Vascular Remodeling: Cytoskeleton Changes and Migration in VSMCs"

_ijms, 2023, doi:10.3390/ijms24032960_

Round 1

Reviewer 1 Report

Dear authors,

I have studied with interest the manuscript «Role of uric acid in vascular remodeling: cytoskeleton changes and migration in VSMCs». The manuscript is well written and the work presented is original.

The authors hypothesized that UA-induced cytoskeleton changes determine an increase in VSMC migratory rate, suggesting UA as a key player in vascular remodeling. The topic is original because searching new mechanisms of UA vascular remodeling as a trigger of cardiovascular diseases is essential.

The results are clearly presented and all the conclusions are supported by the results. All the cited references are relevant to the research and well-balanced. The tables and figure correspond to the description in the text and they are well-designed and reflect important information.

However, I have comments that could improve the quality of the paper:

1.      Please, provide a common scheme of your experiment.

2.      Conclusion should give only the information about the result observed.

Generally, I think that this is a very worthy work. I express my gratitude to the authors for their work and my great pleasure in reading their results.

Reviewer 2 Report

Thank you for the opportunity to review this article titled “Role of uric acid in vascular remodeling: cytoskeleton changes and migration in VSMCs”. This was an interesting read and describes the role of uric acid in vascular dysfunction and remodeling along with implications. It would be interesting to see these concepts applied in a clinical context as well in the future. The article is scientifically sound and well presented. I feel that this paper would be valuable for readers.

The only change I would recommend would be to move the ‘Materials and Methods’ section after the introduction and prior to describing the results.

Round 2

Reviewer 1 Report

The authors have adressed all my comments.